# Sticking the Landing: Simple, Lower-Variance Gradient Estimators for Variational Inference

**Geoffrey Roeder**
University of Toronto
roeder@cs.toronto.edu

**Yuhuai Wu**
University of Toronto
ywu@cs.toronto.edu

**David Duvenaud**
University of Toronto
duvenaud@cs.toronto.edu

## Abstract

We propose a simple and general variant of the standard reparameterized gradient estimator for the variational evidence lower bound. Specifically, we remove a part of the total derivative with respect to the variational parameters that corresponds to the score function. Removing this term produces an unbiased gradient estimator whose variance approaches zero as the approximate posterior approaches the exact posterior. We analyze the behavior of this gradient estimator theoretically and empirically, and generalize it to more complex variational distributions such as mixtures and importance-weighted posteriors.

## 1 Introduction

Recent advances in variational inference have begun to make approximate inference practical in large-scale latent variable models. One of the main recent advances has been the development of variational autoencoders along with the reparameterization trick [Kingma and Welling, 2013, Rezende et al., 2014]. The reparameterization trick is applicable to most continuous latent-variable models, and usually provides lower-variance gradient estimates than the more general REINFORCE gradient estimator [Williams, 1992].

Intuitively, the reparameterization trick provides more informative gradients by exposing the dependence of sampled latent variables $\mathbf{z}$ on variational parameters $\phi$. In contrast, the REINFORCE gradient estimate only depends on the relationship between the density function $\log q_\phi(\mathbf{z}|\mathbf{x}, \phi)$ and its parameters.

Surprisingly, even the reparameterized gradient estimate contains the score function—a special case of the REINFORCE gradient estimator. We show that this term can

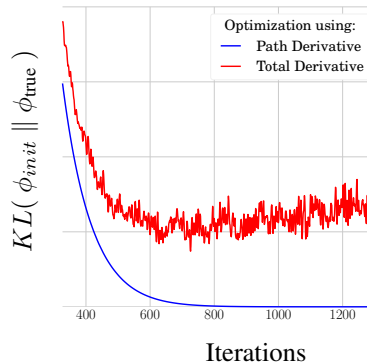

Figure 1: Fitting a 100-dimensional variational posterior to another Gaussian, using standard gradient versus our proposed path derivative gradient estimator.

easily be removed, and that doing so gives even lower-variance gradient estimates in many circumstances. In particular, as the variational posterior approaches the true posterior, this gradient estimator approaches zero variance faster, making stochastic gradient-based optimization converge and "stick" to the true variational parameters, as seen in figure 1.

## 1.1 Contributions

- We present a novel unbiased estimator for the variational evidence lower bound (ELBO) that has zero variance when the variational approximation is exact.
- We provide a simple and general implementation of this trick in terms of a single change to the computation graph operated on by standard automatic differentiation packages.
- We generalize our gradient estimator to mixture and importance-weighted lower bounds, and discuss extensions to flow-based approximate posteriors. This change takes a single function call using automatic differentiation packages.
- We demonstrate the efficacy of this trick through experimental results on MNIST and Omniglot datasets using variational and importance-weighted autoencoders.

## 1.2 Background

Making predictions or computing expectations using latent variable models requires approximating the posterior distribution $p(\mathbf{z}|\mathbf{x})$. Calculating these quantities in turn amounts to using Bayes' rule: $p(\mathbf{z}|\mathbf{x}) = p(\mathbf{x}|\mathbf{z})p(\mathbf{z})/p(\mathbf{x})$.

Variational inference approximates $p(\mathbf{z}|\mathbf{x})$ with a tractable distribution $q_\phi(\mathbf{z}|\mathbf{x})$ parameterized by $\phi$ that is close in KL-divergence to the exact posterior. Minimizing the KL-divergence is equivalent to maximizing the evidence lower bound (ELBO):

$$\mathcal{L}(\phi) = \mathbb{E}_{\mathbf{z} \sim q}[\log p(\mathbf{x}, \mathbf{z}) - \log q_\phi(\mathbf{z} \,|\, \mathbf{x})] \qquad \text{(ELBO)}$$

An unbiased approximation of the gradient of the ELBO allows stochastic gradient descent to scalably learn parametric models. Stochastic gradients of the ELBO can be formed from the REINFORCE-style gradient, which applies to any continuous or discrete model, or a reparameterized gradient, which requires the latent variables to be modeled as continuous. Our variance reduction trick applies to the reparameterized gradient of the evidence lower bound.

## 2 Estimators of the variational lower bound

In this section, we analyze the gradient of the ELBO with respect to the variational parameters to show a source of variance that depends on the complexity of the approximate distribution.

When the joint distribution $p(\mathbf{x}, \mathbf{z})$ can be evaluated by $p(\mathbf{x}|\mathbf{z})$ and $p(\mathbf{z})$ separately, the ELBO can be written in the following three equivalent forms:

$$\mathcal{L}(\phi) = \mathbb{E}_{\mathbf{z} \sim q}[\log p(\mathbf{x}|\mathbf{z}) + \log p(\mathbf{z}) - \log q_\phi(\mathbf{z}|\mathbf{x})] \qquad (1)$$
$$= \mathbb{E}_{\mathbf{z} \sim q}[\log p(\mathbf{x}|\mathbf{z}) + \log p(\mathbf{z}))] + \mathbb{H}[q_\phi] \qquad (2)$$
$$= \mathbb{E}_{\mathbf{z} \sim q}[\log p(\mathbf{x}|\mathbf{z})] - KL(q_\phi(\mathbf{z}|\mathbf{x})||p(\mathbf{z})) \qquad (3)$$

Which ELBO estimator is best? When $p(\mathbf{z})$ and $q_\phi(\mathbf{z}|\mathbf{x})$ are multivariate Gaussians, using equation (3) is appealing because it analytically integrates out terms that would otherwise have to be estimated by Monte Carlo. Intuitively, we might expect that using exact integrals wherever possible will give lower-variance estimators by reducing the number of terms to be estimated by Monte Carlo methods.

Surprisingly, even when analytic forms of the entropy or KL divergence are available, sometimes it is better to use (1) because it will have lower variance. Specifically, this occurs when $q_\phi(\mathbf{z}|\mathbf{x}) = p(\mathbf{z}|\mathbf{x})$, i.e. the variational approximation is exact. Then, the variance of the full Monte Carlo estimator $\hat{\mathcal{L}}_{MC}$ is exactly zero. Its value is a constant, independent of $\mathbf{z} \overset{\text{iid}}{\sim} q_\phi(\mathbf{z}|\mathbf{x})$. This follows from the assumption $q_\phi(\mathbf{z}|\mathbf{x}) = p(\mathbf{z}|\mathbf{x})$:

$$\hat{\mathcal{L}}_{MC}(\phi) = \log p(\mathbf{x}, \mathbf{z}) - \log q_\phi(\mathbf{z}|\mathbf{x}) = \log p(\mathbf{z}|\mathbf{x}) + \log p(\mathbf{x}) - \log p(\mathbf{z}|\mathbf{x}) = \log p(\mathbf{x}), \quad (4)$$

This suggests that using equation (1) should be preferred when we believe that $q_\phi(\mathbf{z}|\mathbf{x}) \approx p(\mathbf{z}|\mathbf{x})$.

Another reason to prefer the ELBO estimator given by equation (1) is that it is the most generally applicable, requiring a closed form only for $q_\phi(\mathbf{z}|\mathbf{x})$. This makes it suitable for highly flexible approximate distributions such as normalizing flows [Jimenez Rezende and Mohamed, 2015], Real NVP [Dinh et al., 2016], or Inverse Autoregressive Flows [Kingma et al., 2016].

**Estimators of the lower bound gradient**  What about estimating the *gradient* of the evidence lower bound? Perhaps surprisingly, the variance of the gradient of the fully Monte Carlo estimator (1) with respect to the variational parameters is not zero, even when the variational parameters exactly capture the true posterior, i.e., $q_\phi(\mathbf{z}|\mathbf{x}) = p(\mathbf{z}|\mathbf{x})$.

This phenomenon can be understood by decomposing the gradient of the evidence lower bound. Using the reparameterization trick, we can express a sample $\mathbf{z}$ from a parametric distribution $q_\phi(\mathbf{z})$ as a deterministic function of a random variable $\epsilon$ with some fixed distribution and the parameters $\phi$ of $q_\phi$, i.e., $\mathbf{z} = t(\epsilon, \phi)$. For example, if $q_\phi$ is a diagonal Gaussian, then for $\epsilon \sim N(0, \mathbb{I})$, $z = \mu + \sigma\epsilon$ is a sample from $q_\phi$.

Under such a parameterization of $\mathbf{z}$, we can decompose the total derivative (TD) of the integrand of estimator (1) w.r.t. the trainable parameters $\phi$ as

$$\hat{\nabla}_{\mathrm{TD}}(\epsilon, \phi) = \nabla_\phi \left[\log p(\mathbf{x}|\mathbf{z}) + \log p(\mathbf{z}) - \log q_\phi(\mathbf{z}|\mathbf{x})\right] \tag{5}$$

$$= \nabla_\phi \left[\log p(\mathbf{z}|\mathbf{x}) + \log p(\mathbf{x}) - \log q_\phi(\mathbf{z}|\mathbf{x})\right] \tag{6}$$

$$= \underbrace{\nabla_\mathbf{z} \left[\log p(\mathbf{z}|\mathbf{x}) - \log q_\phi(\mathbf{z}|\mathbf{x})\right] \nabla_\phi t(\epsilon, \phi)}_{\text{path derivative}} - \underbrace{\nabla_\phi \log q_\phi(\mathbf{z}|\mathbf{x})}_{\text{score function}}, \tag{7}$$

The reparameterized gradient estimator w.r.t. $\phi$ decomposes into two parts. We call these the path derivative and score function components. The path derivative measures dependence on $\phi$ only through the sample $\mathbf{z}$. The score function measures the dependence on $\log q_\phi$ directly, without considering how the sample $\mathbf{z}$ changes as a function of $\phi$.

When $q_\phi(\mathbf{z}|\mathbf{x}) = p(\mathbf{z}|\mathbf{x})$ for all $\mathbf{z}$, the path derivative component of equation (7) is identically zero for all $\mathbf{z}$. However, the score function component is not necessarily zero for any $\mathbf{z}$ in some finite sample, meaning that the total derivative gradient estimator (7) will have nonzero variance even when $q$ matches the exact posterior everywhere.

This variance is induced by the Monte Carlo sampling procedure itself. Figure 3 depicts this phenomenon through the loss surface of $\log p(\mathbf{x}, \mathbf{z}) - \log q_\phi(\mathbf{z}|\mathbf{x})$ for a Mixture of Gaussians approximate and true posterior.

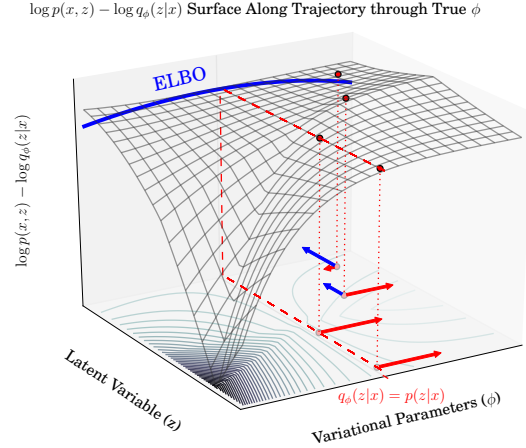

Figure 2: The evidence lower bound is a function of the sampled latent variables $\mathbf{z}$ and the variational parameters $\phi$. As the variational distribution approaches the true posterior, the gradient with respect to the sampled $\mathbf{z}$ (blue) vanishes.

**Path derivative of the ELBO**  Could we remove the high-variance score function term from the gradient estimate? For stochastic gradient descent to converge, we require that our gradient estimate is unbiased. By construction, the gradient estimate given by equation (7) is unbiased. Fortunately, the problematic score function term has expectation zero. If we simply remove that term, we maintain an unbiased estimator of the true gradient:

$$\hat{\nabla}_{\mathrm{PD}}(\epsilon, \phi) = \nabla_\mathbf{z} \left[\log p(\mathbf{z}|\mathbf{x}) - \log q_\phi(\mathbf{z}|\mathbf{x})\right] \nabla_\phi t(\epsilon, \phi) - \cancel{\nabla_\phi \log q_\phi(\mathbf{z}|\mathbf{x})}. \tag{8}$$

This estimator, which we call the path derivative gradient estimator due to its dependence on the gradient flow only through the path variables $\mathbf{z}$ to update $\phi$, is equivalent to the standard gradient estimate with the score function term removed. The path derivative estimator has the desirable property that as $q_\phi(z|x)$ approaches $p(z|x)$, the variance of this estimator goes to zero.

**When to prefer the path derivative estimator**  Does eliminating the score function term from the gradient yield lower variance in all cases? It might seem that its removal can only have a variance reduction effect on the gradient estimator. Interestingly, the variance of the path derivative gradient estimator may actually be higher in some cases. This will be true when the score function is positively correlated with the remaining terms in the total derivative estimator. In this case, the score function acts as a control variate: a zero-expectation term added to an estimator in order to reduce variance.

| **Alg. 1** Standard ELBO Gradient | **Alg. 2** Path Derivative ELBO Gradient |
|---|---|
| **Input:** Variational parameters $\phi_t$, Data $\mathbf{x}$ | **Input:** Variational parameters $\phi_t$, Data $\mathbf{x}$ |
| $\quad \epsilon_t \sim p(\epsilon)$ | $\quad \epsilon_t \sim p(\epsilon)$ |
| $\quad$ **def** $\hat{\mathcal{L}}_t(\phi)$: | $\quad$ **def** $\hat{\mathcal{L}}_t(\phi)$: |
| $\quad\quad \mathbf{z}_t \leftarrow \texttt{sample\_q}(\phi, \epsilon_t)$ | $\quad\quad \mathbf{z}_t \leftarrow \texttt{sample\_q}(\phi, \epsilon_t)$ |
| | $\quad\quad \phi' \leftarrow \texttt{stop\_gradient}(\phi)$ |
| $\quad\quad$ **return** $\log p(\mathbf{x}, \mathbf{z}_t)$ - $\log q(\mathbf{z}_t \vert \mathbf{x}, \phi)$ | $\quad\quad$ **return** $\log p(\mathbf{x}, \mathbf{z}_t)$ - $\log q(\mathbf{z}_t \vert \mathbf{x}, \phi')$ |
| **return** $\nabla_\phi \hat{\mathcal{L}}_t(\phi_t)$ | **return** $\nabla_\phi \hat{\mathcal{L}}_t(\phi_t)$ |

Control variates are usually scaled by an adaptive constant $c^*$, which modifies the magnitude and direction of the control variate to optimally reduce variance, as in Ranganath et al. [2014]. In the preceding discussion, we have shown that $\widehat{c^*} = 1$ is optimal when the variational approximation is exact, since that choice yields analytically zero variance. When the variational approximation is not exact, an estimate of $c^*$ based on the current minibatch will change sign and magnitude depending on the positive or negative correlation of the score function with the path derivative.

Optimal scale estimation procedures is particularly important when the variance of an estimator is so large that convergence is unlikely. However, in the present case of reparameterized gradients, where the variance is already low, estimating a scaling constant introduces another source of variance. Indeed, we can only recover the true optimal scale when the variational approximation is exact in the regime of infinite samples during Monte Carlo integration.

Moreover, the score function must be independently estimated in order to scale it. Estimating the gradient of the score function independent of automatic reverse-mode differentiation can be a challenging engineering task for many flexible approximate posterior distributions such as Normalizing Flows [Jimenez Rezende and Mohamed, 2015], Real NVP [Dinh et al., 2016], or IAF [Kingma et al., 2016].

By contrast, in section 6 we show improved performance on the MNIST and Omniglot density estimation benchmarks by approximating the optimal scale with 1 throughout optimization. This technique is easy to implement using existing automatic differentiation software packages. However, if estimating the score function independently is computationally feasible, and a practitioner has evidence that the variance induced by Monte Carlo integration will reduce the overall variance away from the optimum point, we recommend establishing an annealling schedule for the optimal scaling constant that converges to 1.

## 3 Implementation Details

In this section, we introduce algorithms 1 and 2 in relation to reverse-mode automatic differentiation, and discuss how to implement the new gradient estimator in Theano, Autograd, Torch or Tensorflow Bergstra et al. [2010], Maclaurin et al. [2015], Collobert et al. [2002], Abadi et al. [2015].

Algorithm 1 shows the standard reparameterized gradient for the ELBO. We require three function definitions: `q_sample` to generate a reparameterized sample from the variational approximation, and functions that implement $\log p(\mathbf{x}, \mathbf{z})$ and $\log q(\mathbf{z} \vert \mathbf{x}, \phi)$. Once the loss $\hat{\mathcal{L}}_t$ is defined, we can leverage automatic differentiation to return the standard gradient evaluated at $\phi_t$. This yields equation (7).

Algorithm 2 shows the path derivative gradient for the ELBO. The only difference from algorithm 1 is the application of the `stop_gradient` function to the variational parameters inside $\hat{\mathcal{L}}_t$. Table 1 indicates the names of `stop_gradient` in popular software packages.

| Theano: | `T.gradient.disconnected_grad` |
|---|---|
| Autograd: | `autograd.core.getval` |
| TensorFlow: | `tf.stop_gradient` |
| Torch: | `torch-autograd.util.get_value` |

Table 1: Functions that implement `stop_gradient`

**Alg. 3** Path Derivative Mixture ELBO Gradient

**Input:** Params $\boldsymbol{\pi}_t = \{\boldsymbol{\pi}_t^j\}_{j=1}^K$, $\boldsymbol{\phi}_t = \{\boldsymbol{\phi}_t^i\}_{i=1}^K$, Data $\mathbf{x}$

  $\epsilon_t \sim p(\epsilon)$
  $\boldsymbol{\phi}_t', \boldsymbol{\pi}_t' \leftarrow$ `stop_gradient`$(\boldsymbol{\phi}_t, \boldsymbol{\pi}_t)$
  **def** $\hat{\mathcal{L}}_t^c(\boldsymbol{\phi})$:
    $\mathbf{z}_t^c \leftarrow$ `sample_q`$(\boldsymbol{\phi}, \epsilon_t)$
    **return** $\log p(\mathbf{x}, \mathbf{z}_t^c)$ - $\log \sum_{c=1}^K \boldsymbol{\pi}_t'^c q(\mathbf{z}_t^c|\mathbf{x}, \boldsymbol{\phi}_t')$
  **return** $\nabla_{\boldsymbol{\phi}, \boldsymbol{\pi}} \left( \sum_{c=1}^K \boldsymbol{\pi}_t^c \hat{\mathcal{L}}_t^c(\boldsymbol{\phi}_t^c) \right)$

**Alg. 4** IWAE ELBO Gradient

**Input:** Params $\boldsymbol{\phi}_t$, Data $\mathbf{x}$

  $\epsilon_1, \epsilon_2, \ldots, \epsilon_K \sim p(\epsilon)$
  $\boldsymbol{\phi}_t' \leftarrow$ `stop_gradient`$(\boldsymbol{\phi}_t)$
  **def** $w_i(\boldsymbol{\phi}, \epsilon_i)$:
    $\mathbf{z}_i \leftarrow$ `sample_q`$(\boldsymbol{\phi}, \epsilon_i)$
    **return** $\frac{p(\mathbf{x}, \mathbf{z}_i)}{q(\mathbf{z}_i|\mathbf{x}, \boldsymbol{\phi}_t')}$
  **return** $\nabla_{\boldsymbol{\phi}} \log \left( \frac{1}{k} \sum_{i=1}^K w_i(\boldsymbol{\phi}, \epsilon_i) \right)$

This simple modification to algorithm 1 generates a copy of the parameter variable that is treated as a constant with respect to the computation graph generated for automatic differentiation. The copied variational parameters are used to evaluate variational the density $\log q_{\boldsymbol{\phi}}$ at $\mathbf{z}$.

Recall that the variational parameters $\boldsymbol{\phi}$ are used both to generate $\mathbf{z}$ through some deterministic function of an independent random variable $\epsilon$, and to evaluate the density of $\mathbf{z}$ through $\log q_{\boldsymbol{\phi}}$. By blocking the gradient through variational parameters in the density function, we eliminate the score function term that appears in equation (7). Per-iteration updates to the variational parameters $\boldsymbol{\phi}$ rely on the $\mathbf{z}$ channel only, e.g., the path derivative component of the gradient of the loss function $\hat{\mathcal{L}}_t$. This yields the gradient estimator corresponding to equation (8).

# 4 Extensions to Richer Variational Families

**Mixture Distributions** In this section, we discuss extensions of the path derivative gradient estimator to richer variational approximations to the true posterior.

Using a mixture distribution as an approximate posterior in an otherwise differentiable estimator introduces a problematic, non-differentiable random variable $\boldsymbol{\pi} \sim \text{Cat}(\boldsymbol{\alpha})$. We solve this by integrating out the discrete mixture choice from both the ELBO and the mixture distribution. In this section, we show that such a gradient estimator is unbiased, and introduce an extended algorithm to handle mixture variational families.

For any mixture of K base distributions $q_{\boldsymbol{\phi}}(\mathbf{z}|\mathbf{x})$, a mixture variational family can be defined by $q_{\boldsymbol{\phi}_M}(\mathbf{z}|\mathbf{x}) = \sum_{c=1}^K \boldsymbol{\pi}_c\, q_{\boldsymbol{\phi}_c}(\mathbf{z}|\mathbf{x})$, where $\boldsymbol{\phi}_M = \{\boldsymbol{\pi}_1, ..., \boldsymbol{\pi}_k, \boldsymbol{\phi}_1, ..., \boldsymbol{\phi}_k\}$ are variational parameters, e.g., the weights and distributional parameters for each component. Then, the mixture ELBO $\mathcal{L}_M$ is given by:

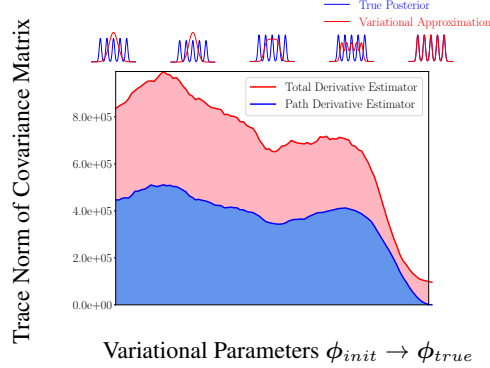

Variational Parameters $\phi_{init} \rightarrow \phi_{true}$

Figure 3: Fitting a mixture of 5 Gaussians as a variational approximation to a posterior that is also a mixture of 5 Gaussians. Path derivative and score function gradient components were measured 1000 times. The path derivative goes to 0 as the variational approximation becomes exact, along an arbitrarily chosen path

$$\sum_{c=1}^K \pi_c \mathbb{E}_{\mathbf{z}_c \sim q_{\phi_c}} \left[ \log p(\mathbf{x}, \mathbf{z}_c) - \log \left( \sum_{k=1}^K \boldsymbol{\pi}_k q_{\phi_k}(\mathbf{z}_c|\mathbf{x}) \right) \right],$$

where the outer sum integrates over the choice of mixture component for each sample from $q_{\boldsymbol{\phi}_M}$, and the inner sum evaluates the density. Applying the new gradient estimator to the mixture ELBO involves applying it to each $q_{\boldsymbol{\phi}_k}(\mathbf{z}_c|\mathbf{x})$ in the inner marginalization.

Algorithm 3 implements the gradient estimator of (8) in the context of a continuous mixture distribution. Like algorithm 2, the new gradient estimator of 3 differs from the vanilla gradient estimator only in the application of `stop_gradient` to the variational parameters. This eliminates the gradient of the score function from the gradient of any mixture distribution.

**Importance-Weighted Autoencoder** We also explore the effect of our new gradient estimator on the IWAE bound Burda et al. [2015], defined as

$$\hat{\mathcal{L}}_K = \mathbb{E}_{\mathbf{z}_1,\ldots,\mathbf{z}_K \sim q(\mathbf{z}|\mathbf{x})} \left[ \log \left( \frac{1}{k} \sum_{i=1}^{K} \frac{p(\mathbf{x}, \mathbf{z}_i)}{q(\mathbf{z}_i|\mathbf{x})} \right) \right] \tag{9}$$

with gradient

$$\nabla_\phi \hat{\mathcal{L}}_K = \mathbb{E}_{\mathbf{z}_1,\ldots,\mathbf{z}_K \sim q(\mathbf{z}|\mathbf{x})} \left[ \sum_{i=1}^{K} \tilde{\mathbf{w}}_i \nabla_\phi \log \mathbf{w}_i \right] \tag{10}$$

where $\mathbf{w}_i := p(\mathbf{x}, \mathbf{z}_i)/q(\mathbf{z}_i|\mathbf{x})$ and $\tilde{\mathbf{w}}_i := \mathbf{w}_i / \sum_{i=1}^{k} \mathbf{w}_i$. Since $\nabla_\phi \log \mathbf{w}_i$ is the same gradient as the Monte Carlo estimator of the ELBO (equation (7)), we can again apply our trick to get a new estimator.

However, it is not obvious whether this new gradient estimator is unbiased. In the unmodified IWAE bound, when $q = p$, the gradient with respect to the variational parameters reduces to:

$$\mathbb{E}_{\mathbf{z}_1,\ldots,\mathbf{z}_k \sim q(\mathbf{z}|\mathbf{x})} \left[ - \sum_{i=1}^{k} \tilde{\mathbf{w}}_i \nabla_\phi \log q_\phi(\mathbf{z}_i|\mathbf{x}) \right]. \tag{11}$$

Each sample $z_i$ is used to evaluate both $\tilde{\mathbf{w}}_i$ and the partial derivative term. Hence, we cannot simply appeal to the linearity of expectation to show that this gradient is 0. Nevertheless, a natural extension of the variance reduction technique in equation (8) is to apply our variance reduction to each importance-weighted gradient sample. See algorithm 4 for how to implement the path derivative estimator in this form.

We present empirical validation of the idea in our experimental results section, which shows markedly improved results using our gradient estimator. We observe a strong improvement in many cases, supporting our conjecture that the gradient estimator is unbiased as in the mixture and multi-sample ELBO cases.

**Flow Distributions** Flow-based approximate posteriors such as Kingma et al. [2016], Dinh et al. [2016], Jimenez Rezende and Mohamed [2015] are a powerful and flexible framework for fitting approximate posterior distributions in variational inference. Flow-based variational inference samples an initial $\mathbf{z}_0$ from a simple base distribution with known density, then learns a chain of invertible, parameterized maps $f_k(\mathbf{z}_{k-1})$ that warp $\mathbf{z}_0$ into $\mathbf{z}_K = f_K \circ f_{K-1} \circ ... \circ f_1(\mathbf{z}_0)$. The endpoint $\mathbf{z}_K$ represents a sample from a more flexible distribution with density $\log q_K(\mathbf{z}_K) = \log q_0(\mathbf{z}_0) - \sum_{k=1}^{K} \log \left| \det \frac{\partial f_k}{\partial \mathbf{z}_{k-1}} \right|$.

We expect our gradient estimator to improve the performance of flow-based stochastic variational inference. However, due to the chain composition used to learn $\mathbf{z}_K$, we cannot straightforwardly apply our trick as described in algorithm 2. This is because each intermediate $\mathbf{z}_j, 1 \leq j \leq K$ contributes to the path derivative component in equation (8). The log-Jacobian terms used in the evaluation of $\log q(\mathbf{z}_k)$, however, require this gradient information to calculate the correct estimator. By applying `stop_gradient` to the variational parameters used to generate each intermediate $\mathbf{z}_i$ and passing only the endpoint $\mathbf{z}_K$ to a log density function, we would lose necessary gradient information at each intermediate step needed for the gradient estimator to be correct. At time of writing, the requisite software engineering to track and expose intermediate steps during backpropagation is not implemented in the packages listed in Table 1, and so we leave this to future work.

## 5 Related Work

Our modification of the standard reparameterized gradient estimate can be interpreted as adding a control variate, and in fact Ranganath et al. [2014] investigated the use of the score function as a control variate in the context of non-reparameterized variational inference. The variance-reduction effect we use to motivate our general gradient estimator has been noted in the special cases of Gaussian distributions with sparse precision matrices and Gaussian copula inference in Tan and Nott [2017] and Han et al. [2016] respectively. In particular, Tan and Nott [2017] observes that by

|  |  | MNIST | | | | Omniglot | | | |
|  |  | VAE | | IWAE | | VAE | | IWAE | |
| stochastic layers | k | Total | Path | Total | Path | Total | Path | Total | Path |
|---|---|---|---|---|---|---|---|---|---|
| 1 | 1 | 86.76 | **86.40** | 86.76 | **86.40** | 108.11 | **107.39** | 108.11 | **107.39** |
|  | 5 | 86.47 | **86.33** | 85.54 | **85.20** | 107.62 | **107.40** | 106.12 | **105.42** |
|  | 50 | **86.35** | *86.48* | 84.78 | **84.45** | 107.80 | **107.42** | 104.67 | **104.16** |
| 2 | 1 | 85.33 | **84.77** | 85.33 | **84.77** | 107.58 | **105.22** | 107.56 | **105.22** |
|  | 5 | 85.01 | **84.68** | 83.89 | **83.57** | 106.31 | **104.87** | 104.79 | **103.59** |
|  | 50 | 84.78 | **84.33** | 82.90 | *83.16* | 106.30 | **105.70** | 103.38 | **102.86** |

Table 2: Results on variational (VAE) and importance-weighted (IWAE) autoencoders using the total derivative estimator, equation (7), versus the path derivative estimator, equation (8) (ours).

eliminating certain terms from a gradient estimator for Gaussian families parameterized by sparse precision matrices, multiple lower-variance unbiased gradient estimators may be derived.

Our work is a generalization to any continuous variational family. This provides a framework for easily implementing the technique in existing software packages that provide automatic differentiation. By expressing the general technique in terms of automatic differentiation, we eliminate the need for case-by-case analysis of the gradient of the variational lower bound as in Tan and Nott [2017] and Han et al. [2016].

An innovation by Ruiz et al. [2016] introduces the generalized reparameterization gradient (GRG) which unifies the REINFORCE-style and reparameterization gradients. GRG employs a weaker form of reparameterization that requires only the first moment to have no dependence on the latent variables, as opposed to complete independence as in Kingma and Welling [2013]. GRG improves on the variance of the score-function gradient estimator in BBVI without the use of Rao-Blackwellization as in Ranganath et al. [2014]. A term in their estimator also behaves like a control variate.

The present study, in contrast, develops a simple drop-in variance reduction technique through an analysis of the functional form of the reparameterized evidence lower bound gradient. Our technique is developed outside of the framework of GRG but can strongly improve the performance of existing algorithms, as demonstrated in section 6. Our technique can be applied alongside GRG.

In the python toolkit Edward [Tran et al., 2016], efforts are ongoing to develop algorithms that implement stochastic variational inference in general as a black-box method. In cases where an analytic form of the entropy or KL-divergence is known, the score function term can be avoided using Edward. This is equivalent to using equations (2) or (3) respectively to estimate the ELBO. As of release 1.2.4 of Edward, the total derivative gradient estimator corresponding to (7) is used for reparameterized stochastic variational inference.

## 6 Experiments

**Experimental Setup**    Because we follow the experimental setup of Burda et al. [2015], we review it briefly here. Both benchmark datasets are composed of $28 \times 28$ binarized images. The MNIST dataset was split into $60,000$ training and $10,000$ test examples. The Omniglot dataset was split into $24,345$ training and $8070$ test examples. Each model used Xavier initialization [Glorot and Bengio, 2010] and trained using Adam with parameters $\beta_1 = 0.9$, $\beta_2 = 0.999$, and $\epsilon = 1e^{-4}$ with 20 observations per minibatch [Kingma and Ba, 2015]. We compared against both architectures reported in Burda et al. [2015]. The first has one stochastic layer with 50 hidden units, encoded using two fully-connected layers of 200 neurons each, using a tanh nonlinearity throughout. The second architecture is two stochastic layers: the first stochastic layer encodes the observations, with two fully-connected layers of 200 hidden units each, into 100 dimensional outputs. The output is used as the parameters of diagonal Gaussian. The second layer takes samples from this Gaussian and passes them through two fully-connected layers of 100 hidden units each into 50 dimensions.

See table 2 for NLL scores estimated as the mean of equation (9) with k=5000 on the test set. We can see that the path derivative gradient estimator improves over the original gradient estimator in all but two cases.

**Benchmark Datasets**   We evaluate our path derivative estimator using two benchmark datasets: MNIST, a dataset of handwritten digits [LeCun et al., 1998], and Omniglot, a dataset of handwritten characters from many different alphabets [Lake, 2014]. To underscore both the easy implementation of this technique and the improvement it offers over existing approaches, we have empirically evaluated our new gradient estimator by a simple modification of existing code[1] [Burda et al., 2015].

**Omniglot Results**   For a two-stochastic-layer VAE using the multi-sample ELBO with gradient corresponding to equation (8) improves over the results in Burda et al. [2015] by 2.36, 1.44, and 0.6 nats for k={1, 5, 50} respectively. For a one-stochastic-layer VAE, the improvements are more modest: 0.72, 0.22, and 0.38 nats lower for k={1, 5, 50} respectively. A VAE with a deep recognition network appears to benefit more from our path derivative estimator than one with a shallow recognition network. For comparison, a VAE using the path derivative estimator with k=5 samples performs only 0.08 nats worse than an IWAE using the total derivative gradient estimator (7) and 5 samples. By contrast, using the total derivative (vanilla) estimator for both models, IWAE otherwise outperforms VAE for k=5 samples by 1.52 nats.

By increasing the accuracy of the ELBO gradient estimator, we may also increase the risk of overfitting. Burda et al. [2015] report that they didn't notice any significant problems with overfitting, as the training log likelihood was usually 2 nats lower than the test log likelihood. With our gradient estimator, we observe only 0.77 nats worse performance for a VAE with k=50 compared to k=5 in the two-layer experiments. IWAE using equation (8) markedly outperforms IWAE using equation (7) on Omniglot. For a 2-layer IWAE, we observe an improvement of 2.34, 1.2, and 0.52 nats for k={1, 5, 50} respectively. For a 1-layer IWAE, the improvements are 0.72, 0.7, and 0.51 for k={1, 5, 50} respectively. Just as in the VAE Omniglot results, a deeper recognition network for an IWAE model benefits more from the improved gradient estimator than a shallow recognition network.

**MNIST Results**   For all but one experiment, a VAE with our path derivative estimator outperforms a vanilla VAE on MNIST data. For k=50 with one stochastic layer, our gradient estimator underperforms a vanilla VAE by 0.13 nats. Interestingly, the training NLL for this run is 86.11, only 0.37 nats different than the test NLL. The similar magnitude of the two numbers suggests that training for longer than Burda et al. [2015] would improve the performance of our gradient estimator. We hypothesize that the worse performance using the path derivative estimator is a consequence of fine-tuning towards the characteristics of the total derivative estimator.

For a two-stochastic-layer VAE on MNIST, the improvements are 0.56, 0.33 and 0.45 for k={1, 5, 50} respectively. In a one-stochastic-layer VAE on MNIST, the improvements are 0.36 and 0.14 for k={1, 5} respectively.

The improvements on IWAE are of a similar magnitude. For k=50 in a two-layer path-derivative IWAE, we perform 0.26 nats worse than with a vanilla IWAE. The training loss for the k=50 run is 82.74, only 0.42 nats different. As in the other failure case, this suggests we have room to improve these results by fine-tuning over our method. For a two stochastic layer IWAE, the improvements are 0.66 and 0.22 for k=1 and 5 respectively. In a one stochastic layer IWAE, the improvements are 0.36, 0.34, and 0.33 for k={1, 5, 50} respectively.

# 7   Conclusions and Future Work

We demonstrated that even when the reparameterization trick is applicable, further reductions in gradient variance are possible. We presented our variance reduction method in a general way by expressing it as a modification of the computation graph used for automatic differentiation. The gain from using our method grows with the complexity of the approximate posterior, making it complementary to the development of non-Gaussian posterior families.

Although the proposed method is specific to variational inference, we suspect that similar unbiased but high-variance terms might exist in other stochastic optimization settings, such as in reinforcement learning, or gradient-based Markov Chain Monte Carlo.

## Footnotes

[1]See `https://github.com/geoffroeder/iwae`

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
