[Reviews · NeurIPS 2017]

Reviewer 1



This paper proposes a new gradient estimator for Variational Bayes with zero variance when the variational distribution is exactly equal to the true distribution of latent variables. The paper is written well and clear. I think it is acceptable after some minor revisions. It will be nicer if the case where the variational distribution is different from the true distribution p(z|x) is analyzed in addition. In that case, the expectation of score function is not zero anymore and therefore the proposed estimator introduces some bias. How adversarial can be the effect of this bias? Also, for the mixture ELBO in line 188, shouldn't there be a variational distribution over \pi_c? I don't see any explicit definition of z_c in the manuscript. How do you choose them in the variational mixture and how does that choice affects reparameterization?

Reviewer 2



This submission proposes a method that often reduces the variance of stochastic gradients for variational inference. With the reparameterization trick, the score function is effectively included in the stochastic gradients. Though it may serve as a control variate, more often it increases the variance of the stochastic gradients. Excluding the score function does not bias the stochastic gradients, and often leads to better empirical results. The topic the paper addresses is important: the reparameterization trick has greatly expanded the applicability of variational inference. Yet there has been little systematic evaluation of the different ways of computing stochastic gradients with the reparametrization trick. In addition to addressing an important topic, this paper presents somewhat surprising results: 1) a Monte Carlo approximation to the entropy term may lead to lower variance overall than using the exact entropy term, 2) the score function adds variance unnecessarily to the stochastic gradients. The paper does not require novel mathematical arguments to make its case, or even much math at all. The empirical results are appreciated, but the improvements appear modest. Overall, the paper is well written. I presume equations 5--7 are all correct, but it would help to have them clarified. Why are we differentiating just the integrand of (1), when the measure of the integrand itself involves the variational parameters? How does equality 7 follow? If it’s just the chain rule, are there notational changes that would make that more clear? Or some additional text to explain it? Notational challenges may be partly why no one has noticed that the score function is adding unnecessary variance to the gradients until now.

Reviewer 3



The authors analyze the functional form of the reparameterization trick for obtaining the gradient of the ELBO, noting that one term depends on the gradient wrt. the sample z, which they call the path derivative, and the other term does not depend on the gradient wrt sample z, which they (and others) call the score function. The score function has zero expectation, but is not necessarily zero at particular samples of z. The authors propose to delete the score function term from the gradient, and show can be done with a one-line modification in popular autodiff software. They claim that this can reduce the variance—in particular, as the variational posterior approaches the true posterior, the variance in the stochastic gradient approaches zero. They also extent this idea nontrivially to more complicated models. While this is a useful and easy-to-use trick, particularly in the way it is implemented in autodiff software, this does not appear to be a sufficiently novel contribution. The results about the score function are well-known, and eliminating or rescaling the score function seems like a natural experiment to do. The discussion of control variates in lines 124-46 is confusing and seemingly contradictory. From my understanding, setting the control variate scale c = 1 corresponds to including the score function term, i.e. using the ordinary ELBO gradient. Their method proposes to not use a control variable, i.e. set c = 0. But lines 140-6 imply that they got better experimental results setting c = 1, i.e. the ordinary ELBO gradient, than with c = 0, i.e. the ELBO gradient without the score function term. This contradicts their experimental results. Perhaps I have misunderstood this section, but I’ve read it several times and still can’t come to a sensible interpretation. The experimental results show only marginal improvements, though they are essentially free.